# Optical Control of the Localized Surface Plasmon Resonance in a Heterotype and Hollow Gold Nanosheet

**DOI:** 10.3390/nano13121826

**Published:** 2023-06-08

**Authors:** Yu Chen, Kai Yin, Yuxuan Xu, Min Liu, Han Huang, Fangping Ouyang

**Affiliations:** 1School of Physics and Electronics and Institution of Super-Microstructure and Ultrafast Process in Advanced Materials, Central South University, 932 Lushannan Road, Changsha 410083, China; 2Powder Metallurgy Research Institution and State Key Laboratory of Powder Metallurgy, Central South University, 932 Lushannan Road, Changsha 410083, China; 3School of Physics and Technology, State Key Laboratory of Chemistry and Utilization of Carbon Based Energy Resources, Xinjiang University, Urumqi 830046, China

**Keywords:** localized surface plasmon resonance, surface-enhanced Raman scattering, LSPR coupling, waveguide switch

## Abstract

The remote excitation and remote-controlling of the localized surface plasmon resonance (LSPR) in a heterotype and hollow gold nanosheet (HGNS) is studied using FDTD simulations. The heterotype HGNS contains an equilateral and hollow triangle in the center of a special hexagon, which forms a so-called hexagon–triangle (H–T) heterotype HGNS. If we focus the incident-exciting laser on one of the vertexes of the center triangle, the LSPR could be achieved among other remote vertexes of the outer hexagon. The LSPR wavelength and peak intensity depend sensitively on factors such as the polarization of the incident light, the size and symmetry of the H–T heterotype structure, etc. Several groups of the optimized parameters were screened out from numerous FDTD calculations, which help to further obtain some significant polar plots of the polarization-dependent LSPR peak intensity with two-petal, four-petal or six-petal patterns. Remarkably, based on these polar plots, the on-off switching of the LSPR coupled among four HGNS hotspots could be remote-controlled simply via only one polarized light, which shows promise for its potential application in remote-controllable surface-enhanced Raman scattering (SERS), optical interconnects and multi-channel waveguide switches.

## 1. Introduction

Surface plasmons (SPs) are the collective oscillation of free electrons which can simultaneously generate evanescent waves propagated along the metal surface [1,2]. When they are excited by light, SPs will be converted into surface plasmon polaritons (SPPs) and are able to be controlled and localized within a special metallic nanostructure in those certain regions such as the tips, the edges or the gaps [3,4,5,6]. The localized surface plasmon (LSP) and LSPR can effectively enhance the localized electromagnetic field (LEF), which is thought to be a novel technique that possesses potential applications in many fields such as SERS, SERS sensors, plasmon-mediated chemical reactions and so on [4,7,8,9,10].

Traditionally, however, the excitation of LSPR as well as the subsequent detection of LSPR scattering share one and the same light channel. As we know, the spectrum of LSPR scattering (or SERS) generally contains abundant and important information [11]. For example, it could contain surface morphology and symmetry information of an LSPR structure itself, and it could also contain the bonding and polarization information of molecules adsorbed on that LSPR structure. The conventional method referred to above is not the best way to excite LSPR or SERS because it will eventually force us to obtain a compounded signal with the background noise of the incident light [12]. This becomes a great obstacle for the application of SERS especially in some special circumstances such as in the biosensing of living cells or in analyzing the polarization characteristics of specific molecules [13,14].

For this reason, a potential alternative method was reported that could remotely excite LSPR and SERS. The method is based on the interactions between LSP and propagated surface plasmon polaritons (which could be named as LSP-PSPP coupling) [15,16] and then the SERS excitation and SERS detection could be worked on two individual paths, which make it possible to obtain a relatively independent and pure scattering spectrum without the interference of the incident light. For example, a SERS remote-excitation device based on silver nanowires (AgNWs) was successfully applied as endoscopic probes for the SERS biosensing of live HeLa cells [13]. The device was able to obtain SERS signals inside of HeLa cells simultaneously without excessive exposure to the SERS exciting laser. The applications for the remote excitation of SERS could also be found in surface catalytic reactions [17], LSPR manipulations [18], remote entanglement of quantum emitters [12], etc. However, the studies introduced above were mainly concentrated on AgNW and AgNW-based systems (for example, a silver nanoparticle-nanowire junction [12,13], branched AgNW dimer [18], etc.). Could this phenomenon be observed in other systems? If so, could we find some new phenomena or potentially interesting applications in those new systems?

Most recently, Chen et al. reported a simple two-step approach [4,19,20,21], which could synthesize two-dimensional HGNSs, as shown in Figure 1d–h. HGNS could grow epitaxially from a solid gold nanosheet (SGNS, as shown in Figure 1a–c and Appendix A) in the liquid phase, and it commonly contains a regular or irregular hollow cavity formed concentrically in the center of itself (see Figure 1d–h). The shapes for the center cavity and the outer solid sheet could be either homotype or heterotype (see Figure 1d–h) [20,21]. In fact, there are more heterotype HGNSs among the reaction outcomes (Figure 1e–h). The outline of these HGNSs is normally hexagonal, including two types: regular hexagon or irregular hexagon. The irregular hexagon has three long sides and three short sides cross-linked alternatively. If it is combined with a triangle cavity in the center, a H–T heterotype HGNS is then achieved (as shown in the sketch maps in Figure 1i–k). In this paper, the studies are focused on the polarization-dependent anisotropic properties of those H–T heterotype HGNSs, because, compared with T–T homotype HGNSs, the symmetry of the structure has been destroyed one step further. This breaking of the symmetry is of great benefit to enhance the optical anisotropic properties of HGNSs, and the latter (optical anisotropy) is the foundation for HGNSs to be utilized as polarization-dependent waveguide switches [21]. Therefore, the remote excitation and controlling of SPPs and LSPR among four coupled hotspots (i.e., the four remote vertexes of the outer hexagon) were systematically evaluated using FDTD simulations.

## 2. Theoretical Method and Computational Model

### 2.1. Definition of the Heterotype Hexagon–Triangle HGNS

The computational model is a heterotype H–T HGNS, which was selected and inspired by the observed samples achieved in our experiments. All HGNSs were synthesized via a simple two-step growth based on SGNSs. Figure 1a–c and Appendix A show some scanning electron microscope (SEM) images of pre-synthesized SGNSs. There are many shapes such as triangle SGNSs (see Figure 1a), irregular hexagon SGNSs (see Figure 1b), regular hexagon SGNSs (see Figure 1c) and so on. Using SGNSs as templates, HGNSs could be synthesized by a further liquid-phase epitaxial growth. Figure 1d–h display some SEM images of several different types of HGNSs. Among these HGNSs, our main focus was on an, H–T HGNS, as shown in Figure 1d and the sketch maps in Figure 1i–k. For simplicity, the center triangle and outer hexagon are concentric, and three sides of the triangle are equivalent with the length fixed in the simulations. The sketch map of the H–T HGNSs and the corresponding structure parameters are displayed in Figure 1i–k. Therein, *D_IN_* represents the length of one side of the center triangle, the value is 1.55 μm and kept unchanged. *D_OL_* and *D_OS_* are used to discriminate the long and short side of the outer and irregular hexagon, respectively. The size of *D_OS_* varies from 0.02 to 0.70 μm with an interval of 0.04 μm, but the perimeter is fixed and unchanged (i.e., the total length of three long sides and three short sides is equal to 6.0 μm: 3*D_OL_* + 3*D_OS_* = 6.0 μm).

The incident-exciting laser is put on one vertex of the center triangle (see Figure 1j), while four monitors, which are used to record and collect the FDTD modeling data, are put on the observation points (OPs) 1–4, respectively, as illustrated in Figure 1j. *θ*_OP-i_ represents the polarization angle of the incident laser, wherein the subscript (OP-i, i = 1, 2, 3, 4) corresponds to OP1~OP4, accordingly. Except for some special orientations, four OPs are not symmetrical with regard to the polarization of the incident light. The LEF at four OPs are different from each other and depend on *θ*_OP-i_. Therefore, we need four independent polarization angles (*θ*_OP-1_~*θ*_OP-4_) to distinguish them. Figure 1k shows a schematic illustration of the remote-controllable LSPR based on H–T heterotype HGNSs.

### 2.2. Computational FDTD Modelling Details

The theoretical method used in this article for solving electromagnetic problems was the finite-difference time-domain (FDTD) method [22,23,24,25]. As first proposed by Yee in 1966 [26], this method is a simple and direct way to discretize the differential form of Maxwell’s equations.

The computational domain was a cuboid with the three-dimensional size of 5.0 × 5.0 × 0.5 μm surrounded by an extra 20 layers of PML-absorbing boundary. In the simulations, the orientation of the object (thin H–T HGNS, only 20 nm in thickness) was vertical to the *xy*-coordinate plane (as illustrated in Appendix A) and in this plane, the HGNS was symmetrical to the *x* direction (See Figure 1j). These settings could reduce the errors resulting from the discretization of non-symmetrical structures by orthogonal FDTD meshes. The mesh size was 2.0 nm and uniform in all directions. The corresponding time step was 3.84 × 10^−18^ s in intervals, which could yield stable and accurate solutions in the wavelength around 0.8–2.2 μm. The total amount of memory required in the FDTD simulations was 8.5 Gigabyte.

The source plane was a Gaussian-shaped pulse propagated along the negative z-direction that was 20 grids (about 40 nm) away from the upper surface of HGNS. The waist radius of the source plane was 0.2 μm. When a polarized incident laser was focused on one vertex of the center triangle, four monitors were placed on the remote vertexes of the outer hexagon to record the FDTD modeling and LEF data, as shown in Figure 1i–k.

The refractive indices for gold nanosheets at the corresponding wavelength (0.8–2.2 μm) were extracted from the Johnson and Christy data [27]. We used the multiple Lorentzian dispersion model to fit the real and imaginary parts of the refractive index. The corresponding fitted curve can be seen in Appendix A.

## 3. Results and Discussion

In a given metallic nanostructure, the LSPR and resonance wavelength are dependent on many factors such as the structure size, symmetry, surface morphology, etc. Among these factors, the structure symmetry is sensitive to the polarization of the exciting light. Therefore, in order to clarify the interactions between the structure symmetry and exciting light polarization and also to clarify how these interactions affect the remote-excited SPPs and LSPR, the polarization-dependent LSPR spectra of H–T HGNS were firstly and systematically calculated. The size of *D_OS_* varies from 0.02 μm to 0.70 μm, whereas the total length of six sides of the outer hexagon is kept constant to 6.0 μm. Thus, *D_OL_* will vary along with the variation in *D_OS_.* Except for a few special cases, the size of the center triangle is also kept unchanged (*D_IN_* = 1.55 μm). It must be pointed out that in order to obtain desirable results in some special cases, we sometimes need to optimize some parameters appropriately. In these cases, *D_IN_* is slightly adjusted according to the practical requirements. Figure 2 and Figure 3 are the polarization-dependent LSPR spectra obtained from OP1, while *D_OS_* is within the range of 0.02–0.34 μm and 0.38–0.70 μm, respectively. The spectra are convenient for us to trace the position of each LSPR peak and its dependence on *θ*_OP-i_ as well as *D_OS_*. When *D_OS_* is smaller than 0.22 μm, four obvious LSPR peaks could be observed around the wavelength of 0.95, 1.10, 1.30 and 1.70 μm, respectively. With the increase in *D_OS_*, the variation in the LSPR peak position is not simply red-shift or blue-shift, because some of the main LSPR peaks will be split into more subpeaks, or conversely, some subpeaks will also be able to merge into one main peak. For example, the peak-splitting begins when *D_OS_* is larger than 0.22 μm for the LSPR peak around the wavelength of 1.30 μm (see Figure 2e–g and Appendix A). When *D_OS_* reaches 0.26 μm, the LSPR peak initially around 1.30 μm has been split into two subpeaks, of which one is centered at 1.172 μm and the other is centered at 1.347 μm (see Figure 2g and Appendix A). On the contrary, when *D_OS_* increases from 0.26 to 0.46 μm, two LSPR subpeaks, which are around 0.913 and 1.042 μm, respectively, will be merged into one peak firstly, and then split apart again (see Figure 2g–i and Figure 3a–c). Therefore, it is difficult to judge the shift of these LSPR peaks when *D_OS_* is just varied within this range (0.26–0.46 μm). The result is not surprising, because although the length of *D_OS_* is increased, the total length of the outer hexagon (6.0 μm) has not been changed, which means that *D_O__L_* will be subsequently decreased.

Interestingly, however, there is an LSPR peak (varied within 1.626–1.719 μm) that has neither peak-splitting nor peak-merging, and it will simply shift to a shorter wavelength with the increase in *D_OS_* in the whole range. In fact, the optical properties of an H–T HGNS can be resolved as the superposing and coupling of several T–T HGNSs. Appendix A is a simple graph which illustrates four types of these resolved T–T HGNSs (as shown in Appendix A respectively) and their variations with the increase in *D_OS_*. It can be inferred from Appendix A that the value of *D_OS_* determines the size, the type (e.g., if it is an equilateral, isosceles or scalene triangle?) and the spatial orientation of the red triangle (as drawn by the red-dashed line in Appendix A). Consequently, that is the reason why the LSPR characteristics will be, finally, determined by *D_OS_*. Among all T–T HGNSs, the type four, which is marked by a blue and dashed-rounded rectangle, is comparatively special and different. It is composed of two equilateral triangles no matter if the size of *D_OS_* is large or small. When *D_OS_* increases, the side length of the red-dashed triangle (type four) decreases monotonically (see the calculation in Appendix A). That might explain why only the blue-shift of the LSPR peak (varied from 1.719 μm to 1.626 μm) was observed with the increase in *D_OS_*. If *D_OS_* is increased to 0.24 μm, it is a critical state. If *D_OS_* is smaller than this value, the blue solid-line triangle is still fully confined within the red dashed-line triangle, or if not, three corners of the blue solid-line triangle will appear at the outside of the red dashed-line triangle. That might be the reason why some LSPR peaks (e.g., the peak around 1.30 μm) begin to split when *D_OS_* reaches 0.24 μm (see Figure 2 and Appendix A).

In addition, the difference in the spatial orientation between two cooperated triangles (e.g., red dashed-line and blue solid-line triangle in Appendix A) will also be changed monotonically along with the increase in *D_OS_* (see Appendix A). These changes in spatial orientation will be reflected in their polarization-dependent LSPR scattering spectra. Figure 4 provides a series of polar plots of the peak intensity as a function of *θ*_OP-1_ in the wavelength range from 1.628 to 1.712 μm. The symbol of _(_*_λ_*_)_*θ*_OP-i(*max*)_ is used to represent the polarization angle of the incident-excited light with regard to the counterpart LSPR maximum. Thus, _(_*_λ_*_)_*θ*_OP-i(*max*)_ is a special *θ*_OP-i_, with which the excited LEF obtains its maximum value because of LSPR. In the symbol of _(_*_λ_*_)_*θ*_OP-i(*max*)_, the left subscript “*λ*” represents the concerned LSPR wavelength, whereas the right subscript “OP-i” represents where the signals are collected (e.g., OP1–OP4), and the symbol “*max”* indicates that the data recorded are at a polarization angle (*θ*) when the strongest LEF is obtained. Figure 4 clearly demonstrates the _(_*_λ_*_)_*θ*_OP-i(*max*)_ and its shift tendency with the corresponding LSPR peak when *D_OS_* increases from 0.06 to 0.50 μm. The _(_*_λ_*_)_*θ*_OP-1(*max*)_ varies within a broad range from 140° to 60° (see Figure 4a–l) and the LSPR peak position moved from 1.712 to 1.628 μm monotonically (see Figure 4m). Similar results can be found in Figure 5, Appendix A for the LSPR peaks around 1.10, 1.30 and 2.10 μm, respectively. The differences are mainly regarding how many degrees _(_*_λ_*_)_*θ*_OP-1(*max*)_ will be rotated over and which direction the LSPR peak will move toward with the increase in *D_OS_*. For example, as shown in Figure 5, _(_*_λ_*_)_*θ*_OP-1(*max*)_ for the LSPR peak near 1.30 μm only varies within a very narrow range (from 10° to 30°). The corresponding LSPR wavelength moves from 1.295 to 1.358 μm, the direction of which is just opposite to that in Figure 4. The polarization of the LSPR peak around 1.10 μm will be affected by peak-splitting when *D_OS_* is larger than 0.26 μm. Moreover, as for the LSPR peak around 2.10 μm, LSPR is only manifest when *D_OS_* is larger than 0.38 μm (as shown in Figure 2, Figure 3, Appendix A).

In conclusion, as we mentioned at the beginning, the optical properties of an H–T HGNS can be resolved as the superposing and coupling of several T–T HGNSs. With the variation in *D_OS_*, the structure and symmetry of the resolved T–T HGNSs (as shown in Figure 4 and Appendix A) will thus be modified accordingly. The optical properties, including the propagation and interaction of SPPs, the remote excitation of the LSPR and the polarization characteristics, are closely connected with these two factors (i.e., the structure size and its symmetry). The polarization-dependent LSPR spectra calculated using the FDTD method are able to reflect the details of these influences and interactions. Then, the polar plot of the LSPR peak intensity, which is further extracted from those polarization-dependent LSPR spectra, is the basis for our following research on the remote-controlling of LSPR coupling among four objective OPs.

In the above discussions, the main problems are how the polarization of the incident light affects the remote-excited LSPR at one hotspot (e.g., at OP1). As we know, LSPs could interact with PSPPs to form an LSP–PSPP interaction. Then, the LSPR at different hotspots (e.g., four OPs) could be controlled because of those LSP–PSPP interactions. Therefore, in order to investigate the remote-controlling of the LSPR at different OPs, the remote-excited and polarization-dependent LSPR spectra at OP2 were calculated and compared with that calculated at OP1. Figure 6 displays the compared results for *D_OS_* = 0.26 μm, *D_OS_* = 0.30 μm and *D_OS_* = 0.34 μm, respectively. The results indicate that there is no significant difference between OP1 and OP2 in the position of their corresponding LSPR peaks (the difference is less than 15 nm as compared with Figure 6a–f). However, the difference has a trend to be enlarged with the increase in *D_OS_*. The difference is also relevant to the LSPR wavelength (the shorter the wavelength is, the bigger the difference will be). Figure 6 also displays a comparison between _(_*_λ_*_)_*θ*_OP-1(*max*)_ and _(_*_λ_*_)_*θ*_OP-2(*max*)_ with respect to the corresponding LSPR peaks. Although there is little difference in the LSPR peak position, _(_*_λ_*_)_*θ*_OP-1(*max*)_ and _(_*_λ_*_)_*θ*_OP-2(*max*)_ are quite different and their difference will also be enlarged with the increase in *D_OS_*. The anisotropy of _(_*_λ_*_)_*θ*_OP-i(*max*)_ among those asymmetric OPs at the same LSPR wavelength means that H–T HGNS have a promising potential to be applied in the remote-controlling of LSPR or SERS, remote-controllable waveguide switches or interactions and so on. Notably, two interesting and useful results were found which are highlighted in yellow in Figure 6. They share two common features.

***First***, one of _(_*_λ_*_)_*θ*_OP-i(*max*)_ is equal to 90°, which means that considering the structure symmetry, as for two symmetrical OPs, the polar plot of the peak intensity as a function of *θ*_OP-i_ will be almost overlapped. For example, OP1 and OP4 (also OP2 and OP3) are symmetrical with respect to the *x*-axis (see Figure 1j). With respect to the same axis, the polar plot for *θ*_OP-1_ will be symmetrical to that of *θ*_OP-4_ as well. Thus, when _(_*_λ_*_)_*θ*_OP-i(*max*)_ is equal to 90°, the polarization curves in the polar plot for two corresponding and symmetric OPs will be overlapped and their polarization properties will be mirrored with the *x*-axis. There are two polarization angles equal to 90°, respectively, in Figure 6d,g. The first one is relevant to OP2 (as shown in Figure 6d, *D_OS_* = 0.26 μm, *λ* = 1.347 μm), while the second is relevant to OP1 (as shown in Figure 6g, *D_OS_* = 0.30 μm, *λ* = 1.656 μm).

***Second***, the absolute difference of the two polarization angles is equal to 60° (θ(λ)OP−1(max)−θ(λ)OP−2(max)=60°), as shown in Figure 6d,g. This property will make it possible to obtain a six-petal patterned polar plot for OP1–OP4 and then divide 360° into six equal parts (as shown in Figure 7a). This six-petal polar plot could help us to pick out the required polarization angle (*θ*), with which the LSPR and interactions among three OPs (e.g., three-point coupling, as shown in Figure 7c,d) could be remotely excited and controlled using only one polarized exciting laser.

Based on the above discussions, there are three important parameters that should be firstly ascertained. They are the polarization angle (*θ*) of the incident light, the resonance wavelength (*λ*) that has appropriate polarity with respect to *θ*_OP-i_ (i = 1, 2, 3, 4) and the corresponding *D_OS_* for the above *θ* and *λ*. Figure 6 has provided two groups of these parameters that can be used directly and require no further optimization. The first is when *λ* = 1.347 μm and *D_OS_* = 0.26 μm. Figure 7a shows a six-petal polar plot for the peak intensity at *λ* = 1.347 μm as a function of *θ*_OP-i_ (i = 1, 2, 3, 4), from which the parameter of *θ* for the three-point coupling of the LSPR could thus be determined. As shown in Figure 7b–d, the incident-excited light with three polarization angles (*θ* = 0°, 59° and 121°) was chosen, with which two-point and three-point coupling of the LSPR could be remote-controlled. When *θ* = 0°, LSPR only occurs at OP1 and OP4 (two-point coupling). When *θ* = 59° (or *θ* = 121°), LSPR occurs at three points of OP1, OP2 and OP3 (or OP2, OP3 and OP4, three-point coupling), respectively. The intensity ratio (IR) is defined to compare the LEF intensity at different OPs. The higher IR value means the better performance in optical anisotropy. For example, in Figure 8c,d, the IR is equal to 8.4 which means the intensity of the LEF at the “hotspots” is 8.4 times higher than that at the “cold spots”. The results indicate that the on-off switching of the LSPR could be controlled using a remote technique. The following work, then, is to screen out more and appropriate groups of parameters so as to cover all modes of the control logic. For example, the second group of the practical parameters could be derived from Figure 6i, while *λ* = 1.656 μm and *D_OS_* = 0.30 μm, as shown in Figure 8. The results of Figure 8 are symmetrical and complementary with those of Figure 7. Here, two-point coupling is transferred to OP2 and OP3, and three-point coupling is transferred to OP1, OP3 and OP4 (or OP1, OP2 and OP4), accordingly. All modes, which control the LSPR communicated in three out of four OPs, are completely included because there are only four modes in three-point coupling (i.e., OP1–OP2–OP3, OP1–OP2–OP4, OP1–OP3–OP4 and OP2–OP3–OP4). Figure 8e,f display the mapping of Poynting vectors when *θ =* 0° and *θ =* 64°, respectively. It could help to analyze the flow of electromagnetic power and the transmission of SPPs.

Theoretically, there are six modes in two-point coupling; however, Figure 7 and Figure 8 only display two of them. Figure 9 displays a polar plot which owns four petals, and every two adjacent petals are almost vertical. This property makes it possible to control the LSPR which occurred simultaneously at two out of the four OPs and fill in the blank for those modes of two-point coupling. In order to obtain the desirable results, the parameters in Figure 9a are optimized from Figure 2i and Figure 6i, while *λ* = 1.143 μm and *D_OS_* = 0.34 μm but the size of *D_IN_* is slightly modified from the initial *D_IN_* = 1.55 μm to *D_IN_* = 1.48 μm. When the polarization angle is 43° (or 137°), LSPR could occur simultaneously in OP2–OP4 (or OP1–OP3). Furthermore, the last two modes for two-point coupling (OP3–OP4 and OP1–OP2) can be found in Figure 9e–f. This group of parameters is *λ* = 1.775 μm, *D_OS_* = 0.10 μm and *D_IN_* = 1.69 μm. Therefore, in addition to three-point coupling, all modes for two-point coupling are also completely identified in the H–T HGNS.

Finally, a two-petal patterned polar plot was achieved while *λ* = 1.319 μm, *D_OS_* = 0.10 μm and *D_IN_* = 1.69 μm. As shown in Figure 10, the main feature of this two-petal plot is that four curves for all OPs are almost overlapped. Thus, the intensity of the LEF at OP1–OP4 is almost equivalent no matter how many degrees the polarization angle (*θ*) is. Figure 10b shows the dependence between the LEF intensity and incident polarization (*θ*). The result means that the LEF intensity at four OPs could be controlled and regulated at the same time by tuning *θ*. When *θ* = 0°, the LEF intensity obtains its maximum value and LSPR turns on simultaneously at all OPs (as shown in Figure 10c). However, when *θ* = 90°, four-point on-off switching of LSPR was then obtained (see Figure 10d). Therefore, in addition to two-point coupling and three-point coupling, a group of parameters is now discovered which is suitable for four-point on/off switching of LSPR.

It must be pointed out that the parameters displayed in the main text for each situation are not the only ones. Take two-point coupling; for example, Appendix A shows more groups of parameters which are also suitable for two-point coupling. Under the different conditions, the calculated value of the IR is also different. Thus, if we want to obtain an ideal two-point coupling, three basic features of its polar plot must be satisfied simultaneously. First, as for two-point coupling, four-petal or a quasi-four-petal patterned polar plot is required. Commonly, the polar plot should have eight petals because they are the polarization-dependent characteristics for four points. Under given conditions, the polarization curve for some OPs will be overlapped or enclosed within those for other OPs. A four-petal polar plot is a fit for two-point coupling because there are only four polarized directions, and each direction contains two petals (overlapped or partially overlapped). Second, every two adjacent polarized directions are best to be perpendicular to each other. Third, the minimum value of the polarization curve in the polar plot should be as small as possible. The polar plot in Figure 9a well satisfies the first and second conditions, but does not satisfy the last one. The value of the IR at some OPs is too small (e.g., the minimum IR is only 4.7 in Figure 9c) and the result is thus not ideal. Finally, Appendix A displays three groups of parameters which are able to switch on/off the LSPR only at one OP. As mentioned above, the difference in the LSPR wavelength between OP1 and OP2 (or OP3 and OP4) has a trend to be enlarged with the increase in *D_OS_*. Therefore, it exists as a probability while the position of the LSPR peak for OP1 is just overlapped with the valley for OP2 (as shown in Appendix A). In this case, although it is still a four-petal polar plot, it is not a fit for two-point coupling but is useful for one-point switching.

## 4. Summary

In summary, the optical control of LSPR in an H–T heterotype HGNS is systematically studied using FDTD simulations. In a wide range of *D_OS_* variations (from 0.02 μm to 0.70 μm), we firstly calculated the polarization-dependent LSPR spectra and analyzed the possible mechanisms regarding how the variation in *D_OS_* affects the LSPR, including the LSPR peak position, peak-splitting, peak-merging and so on. Then, at each *D_OS_*, the polarization characteristics of those main LSPR peaks and their peak intensities that depend on the polarization angle (*θ*) were discussed, which helped to further ascertain the variation law and tendency of _(_*_λ_*_)_*θ*_OP-i(*max*)_. Based on that, several groups of optimized parameters were screened out from numerous FDTD calculations, with which two-petal, four-petal and six-petal patterned polar plots of the polarization-dependent LSPR peak intensity were thus obtained. These polar plots, finally, helped to accomplish our initial goals, that is the remote-controlled on-off switching of LSPR and LSPR coupling, including two-point coupling, three-point coupling, etc.

## Figures and Tables

**Figure 1 nanomaterials-13-01826-f001:**
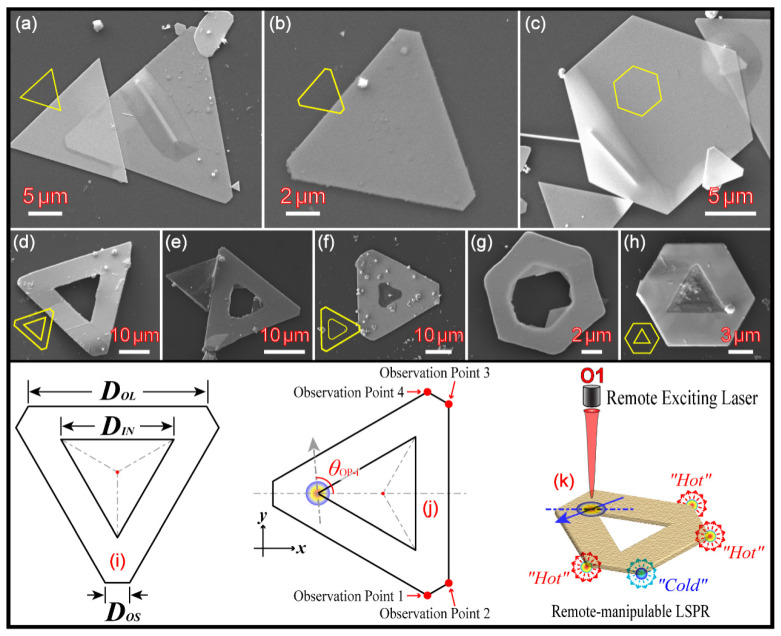
Morphology and structure characterization of SGNSs and HGNSs. (**a**–**c**) SEM images of SGNSs. (**d**–**h**) SEM images of different types of HGNSs. The inset of (**a**–**h**) is the simple sketch of the corresponding SEM image. (**i**,**j**) Sketch map of the computational model and structure parameters. (**k**) Illustration of the remote excitation and remote-controlling of LSPR.

**Figure 2 nanomaterials-13-01826-f002:**
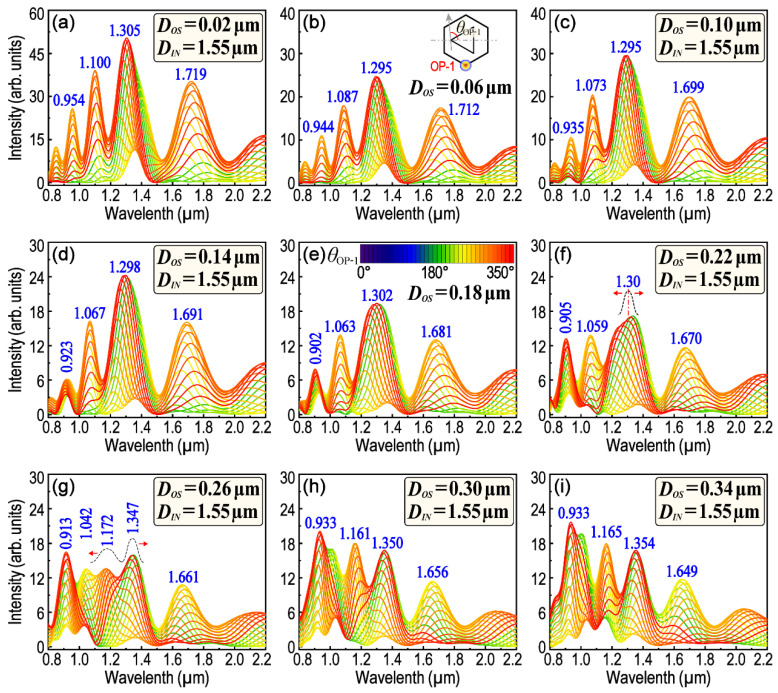
FDTD-calculated polarization-dependent LSPR scattering spectra. (**a**–**i**) *D_OS_* varies from 0.02 μm to 0.34 μm, accordingly. At each *D_OS_*, the spectra were obtained from OP1 (as an inset of (**b**)) and calculated sequentially when *θ*_OP-1_ rotated from 0° to 350° at an interval of 10°. The inset of (**e**) illustrates the the corresponding relationship of *θ*_OP-1_ and each color line.

**Figure 3 nanomaterials-13-01826-f003:**
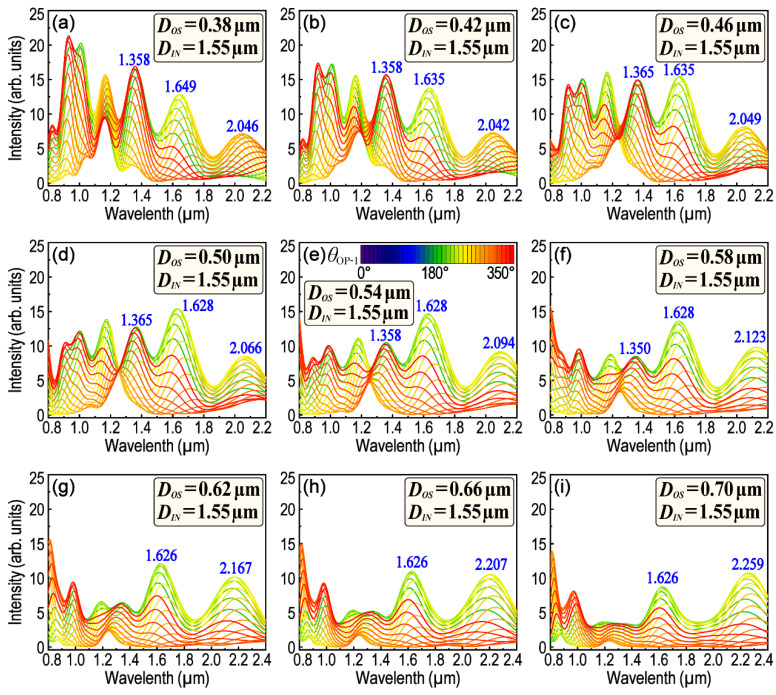
A comparison of the polarization-dependent LSPR scattering spectra while *D_OS_* is within the range from 0.38 μm to 0.70 μm. Rounded rectangle inserted in (**a**–**i**) displays the simulation parameters of *D_OS_* and *D_IN_*.

**Figure 4 nanomaterials-13-01826-f004:**
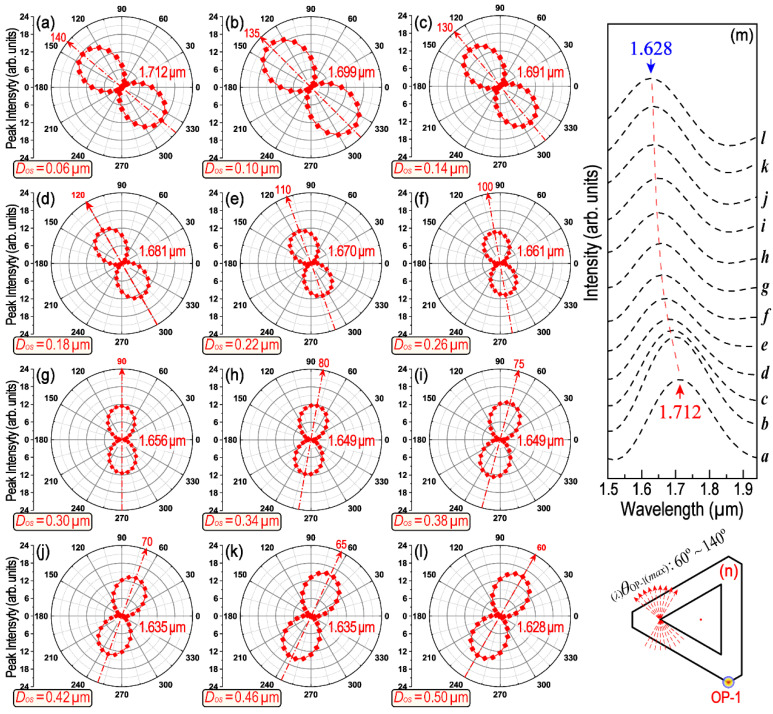
A comparison of the polarization-dependent intensity for the LSPR peak around the wavelength 1.628–1.712 μm. (**a**–**l**) Peak intensity as a function of polarization difference at each *D_OS_*. *D_OS_* covers the range from 0.06 to 0.50 μm. (**m**) Shift of the LSPR peak wavelength at their respective _(_*_λ_*_)_*θ*_OP-1(*max*)_. Italic small letters *a*-*l* correspond to (**a**–**l**). (**n**) Illustration of the covered range of _(_*_λ_*_)_*θ*_OP-1(*max*)_ with the variation in *D_OS_*.

**Figure 5 nanomaterials-13-01826-f005:**
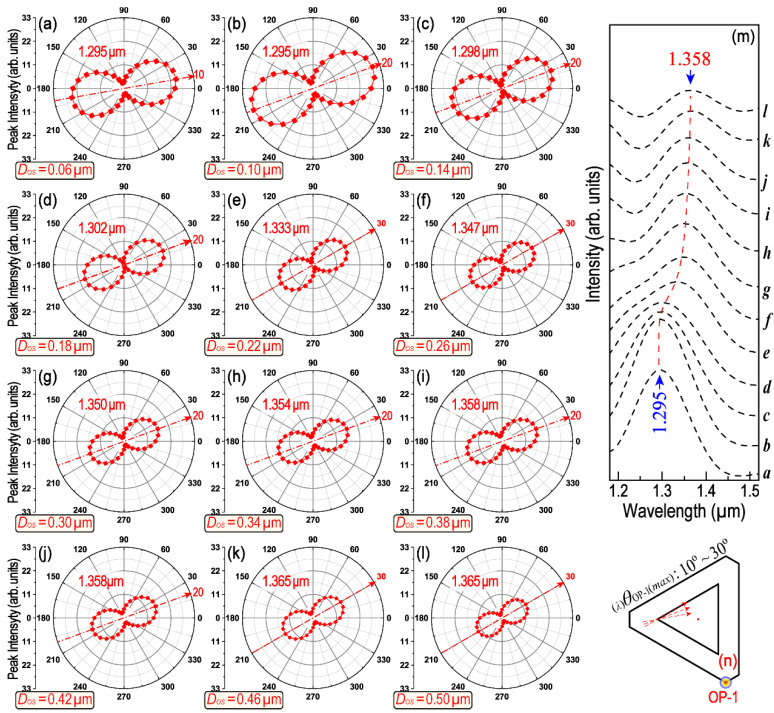
LSPR peak intensity with respect to incident polarization dependence. (**a**–**l**) Rotation tendency of _(_*_λ_*_)_*θ*_OP-1(*max*)_ at different *D_OS_* ranged from 0.06 to 0.50 μm. (**m**) Shift tendency of the LSPR peak corresponds to (**a**–**l**) at their respective _(_*_λ_*_)_*θ*_OP-1(*max*)_. Italic small letters *a*-*l* correspond to (**a**–**l**). (**n**) The range of _(_*_λ_*_)_*θ*_OP-1(*max*)_ with the variation in *D_OS_*.

**Figure 6 nanomaterials-13-01826-f006:**
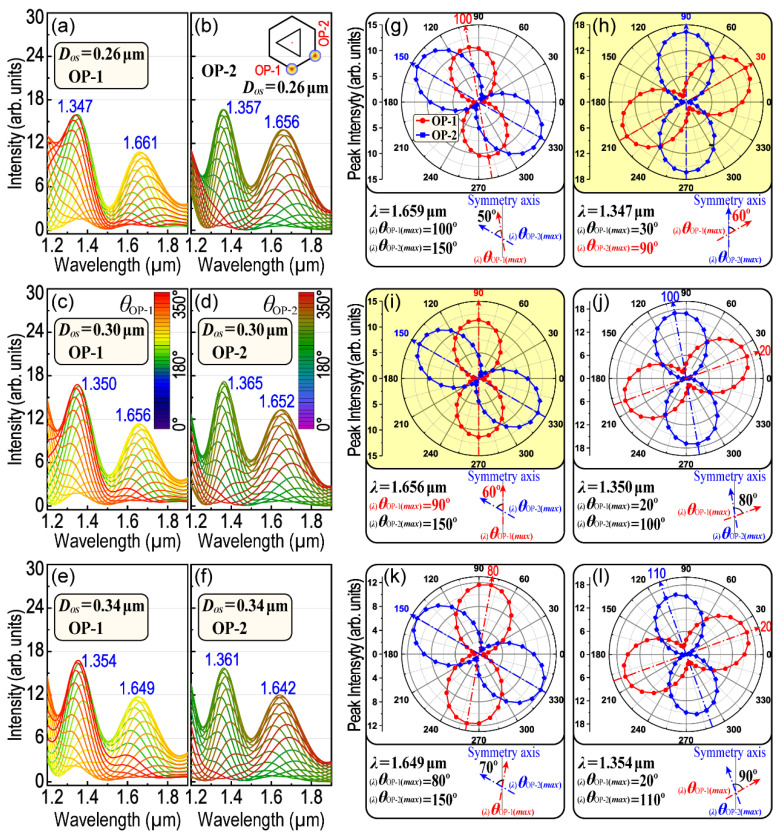
Polarization-dependent anisotropic property of LSPR between OP1 and OP2. Comparison of the polarization-dependent LSPR spectra for OP1 and OP2, while *D_OS_* are 0.26 μm in (**a**,**b**), 0.30 μm in (**c**,**d**) and 0.34 μm in (**e**,**f**). (**g**–**l**); Comparison of the polarization-dependent LSPR peak intensity for OP1 and OP2 correspond to (**a**–**f**).

**Figure 7 nanomaterials-13-01826-f007:**
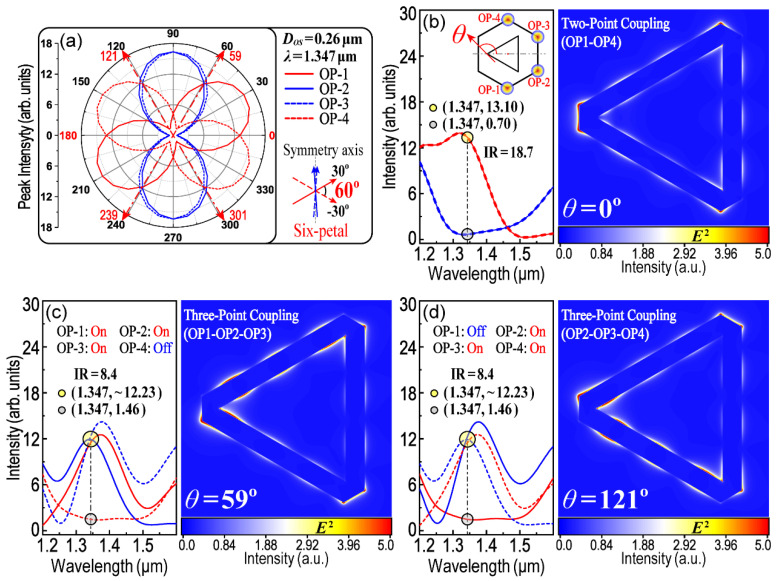
Remote-controlling of LSPR among OP1–OP4. (**a**) A six-petal polar plot. *D_OS_* = 0.26 μm, *λ* = 1.347 μm; (**b**) Two-point coupling between OP1 and OP4; (**c**,**d**) Three-point coupling in OP1–OP2–OP3 and OP2–OP3–OP4.

**Figure 8 nanomaterials-13-01826-f008:**
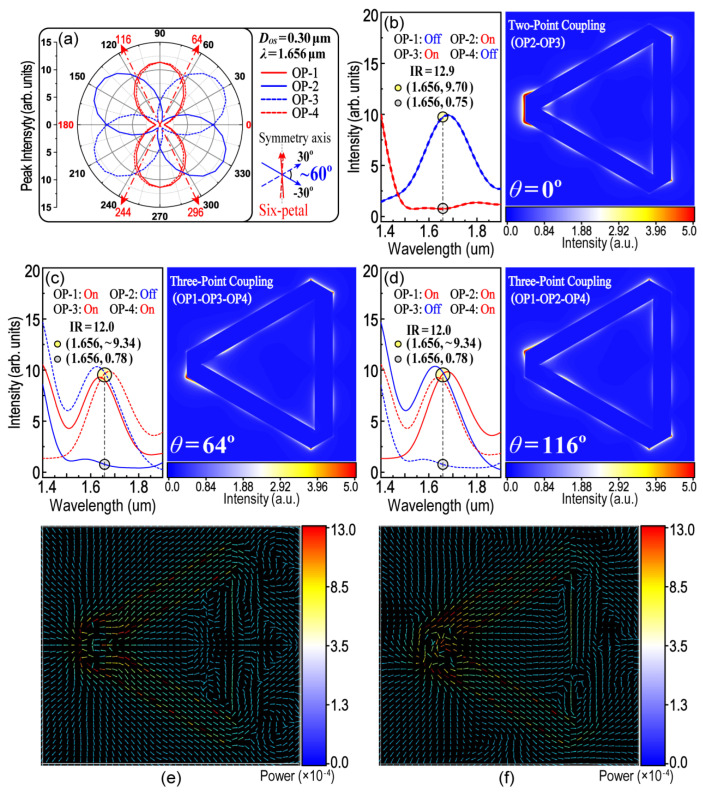
Symmetrical and complementary results with those in Figure 7. (**a**) A six-petal patterned polar plot. *D_OS_* = 0.30 μm, *λ* = 1.656 μm; (**b**) Two-point coupling between OP2 and OP3; (**c**,**d**) Three-point coupling in OP1–OP3–OP4 and OP1–OP2–OP4; (**e**,**f**) Mapping of Poynting vectors (*θ =* 0° and *θ =* 64°).

**Figure 9 nanomaterials-13-01826-f009:**
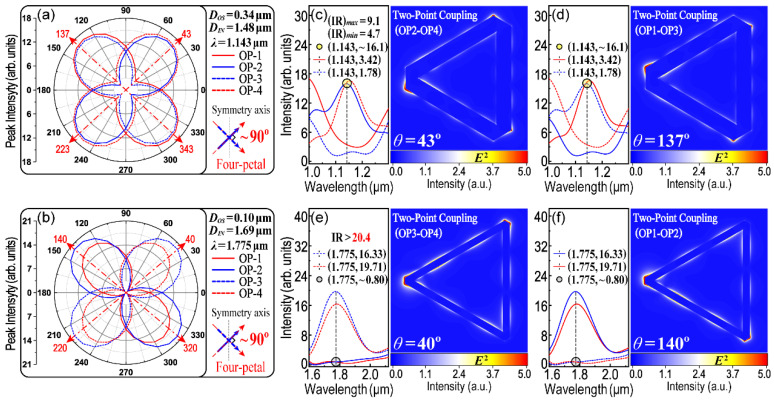
The on-off switching of LSPR between two OPs. Four-petal patterned polar plots of (**a**) *D_OS_* = 0.34 μm, *D_IN_* = 1.48 μm, *λ* = 1.143 μm and (**b**) *D_OS_* = 0.10 μm, *D_IN_* = 1.69 μm, *λ* = 1.775 μm; two-point coupling and its electric field map in (**c**) OP2–OP4, (**d**) OP1–OP3, (**e**) OP3–OP4 and (**f**) OP1–OP2.

**Figure 10 nanomaterials-13-01826-f010:**
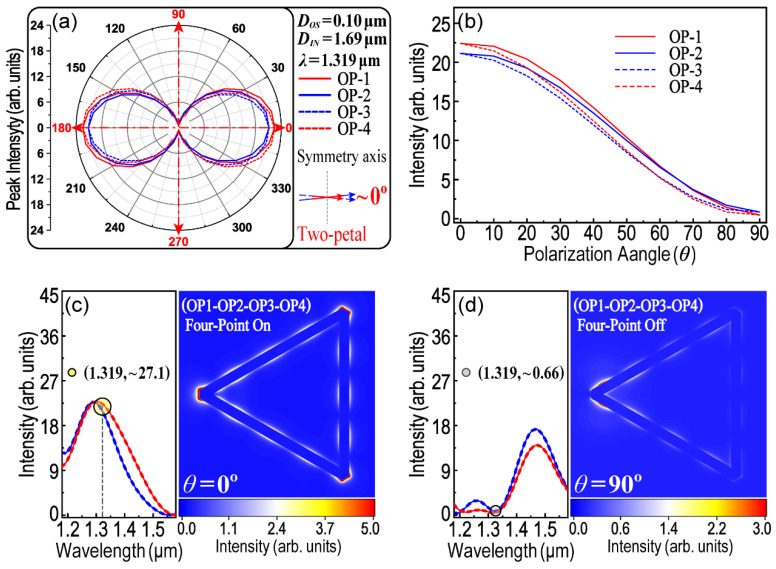
Optical properties of a two-petal polarization. (**a**) Two-petal patterned polar plot. *D_OS_* = 0.10 μm, *D_IN_* = 1.69 μm, *λ* = 1.391 μm; (**b**) Angle-resolved LEF intensity; (**c**) LSPR Four-point On; (**d**) LSPR Four-point Off.

## Data Availability

The raw data are not publicly available at this time but may be obtained from the authors upon reasonable request.

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
