# Peer review of "Optical Control of the Localized Surface Plasmon Resonance in a Heterotype and Hollow Gold Nanosheet"

_nanomaterials, 2023, doi:10.3390/nano13121826_

Round 1

Reviewer 1 Report

The paper presents results of multiple and extensive numerical modeling obtained by FDTD method. Unfortunately, they are not directly related to possible experimental investigation of the similar structures. The problem is in the simulation design that uses the optical launch field with the aperture 200 nm that is far below the diffraction limit. In my understanding the current results could be used as additional to the correct experimentally-related simulations.

Thus I have to recommend the major revision of the paper.

For this revision I decide to provide several recommendations.

  1. One need to use the launch field as the plane wave or a pulsed Gaussian beam with the waist radius of about 5 μm (depending on experimental setup). The additional power monitors could be positions near the boundary of the simulation region in order to measure the reflected and transmitted waves that are the main for the numerical analysis.
  2. For the pulsed FDTD method one need to take into account the dispersion thus a gold nanosheets has to be fitted by the Multiple Lorentzian dispersion model.
  3. The power monitors are described not only by the position but also the orientation that measure the power flow. Thus this information must be discussed and displayed on Fig.1S.
  4. I propose that new data of reflected and transmitted power will contain multiple peeks. Thus the results of the current papar may help to understand the nature of the different peaks. But due to a small launch aperture (200 nm) the results of FDTD simulation may strongly depend on the distence of the launch plate from the structure. Thus it in needed to accomplish additional simulations for different distances from the structure surface and to prove that all presented data are correct.
  5.  Supporting Information is done “Only for reviewers” if so, one need to correct the text which has not any relation to this Supporting Information.

In my understanding this revision is too large to be realized in a short time, thus may be the authors have to withdrew this paper in order to resubmit it later.

PS. It is also needs to correct the misprint (Frist to First).

Author Response

Response to Reviewer’s comments (ID: nanomaterials-2422324):

Thank you greatly for your time and effort spent in judging this work. We have carefully revised the manuscript according to the Reviewers’ comments. The revised parts have been marked in red in the new version.

Response to the reviewer 1:

  1. One need to use the launch field as the plane wave or a pulsed Gaussian beam with the waist radius of about 5 μm (depending on experimental setup). The additional power monitors could be positions near the boundary of the simulation region in order to measure the reflected and transmitted waves that are the main for the numerical analysis.

R: It is a good advice to set the waist radius larger than 5 μm. In fact, at the beginning of our simulations, we have thought about of this problem. But after the following calculation and comparison, this value will only influence the relative intensity of LSPR peak but will not influence the LSPR wavelength (peak position). As show in Figure 1. it shows a comparison of the LSPR scattering spectrum when the waist radius are 2 μm, 5 μm and 8 μm respectively. The results support this point.

Figure 2 shows a xy-distribution of electrical field in the z=0 plane. it can be inferred that there are no obvious reflections on the boundary. The total computational domain is surrounded by over 20 layers of PML absorbing boundary which could avoid evident noise caused by the reflections on the boundary.

Figure 1. A comparison of the LSPR scattering spectrum with different waist radius and z-position of the exciting incident light.

Figure 2. 2D-mapping of the electrical field in the z = 0 plane.

  1. For the pulsed FDTD method one need to take into account the dispersion thus a gold nanosheets has to be fitted by the Multiple Lorentzian dispersion model.

R: In the simulation, we used exactly the Multiple Lorentzian dispersion model to fit the refractive indices for gold nanosheets. The discrete and experimental measured real and imaginary parts of the refractive index were inferred from the Johnson and Christy data.

We have modified the corresponding descriptions.

“The refractive indices for gold nanosheets at the corresponding wavelength (0.8 - 2.2 μm) were extracted from the Ref. 24 [24]. We use the Multiple Lorentzian dispersion model to fit the real and imaginary parts of the refractive index. The corresponding fitted curve was shown in Figs. S1(b) and (c).”

  1. The power monitors are described not only by the position but also the orientation that measure the power flow. Thus this information must be discussed and displayed on Fig.1S.

R: The main point in this article is focused on how dose LSPR and LSPR-coupling could be controlled remotely using one polarized light. We care about the controlling and the enhancement of the localized electrical field. Thus, the monitors (e.g. OPs in the article) only recorded the localized electrical field.

  1. I propose that new data of reflected and transmitted power will contain multiple peeks. Thus the results of the current papar may help to understand the nature of the different peaks. But due to a small launch aperture (200 nm) the results of FDTD simulation may strongly depend on the distence of the launch plate from the structure. Thus it in needed to accomplish additional simulations for different distances from the structure surface and to prove that all presented data are correct.

R: Yes, this is a very good suggestion. In fact, according to our simulations, the interactions between six points at the corner of outer hexagon are quiet complicated. There are too many points which means too many possibilities and combinations. Also, present heterotype hollow Au nanosheet is the composition of a irregular hexagon with a regular triangle, which further complicates the problem. Figure 3 shows a possible interaction mechanism for special LSPR wavelength (around 1.3 μm). We also modified Figure 8 in the main text and added two mappings of Poynting vectors.

In our opinion, this article on the whole has displayed the potential that how to use one polarized light to control LSPR and LSPR-coupling in a remote technique and found the optimized parameters that cover all coupling modes to light on/off of LSPR. Thus, in our following researches, we will do our best and try to clarify the nature for other LSPR peaks.

Figure 3. Possible interaction mechanism for LSPR wavelength around 1.30 μm.

Figure 4. Modified Figure 8 in the main text.

  1. Supporting Information is done “Only for reviewers” if so, one need to correct the text which has not any relation to this Supporting Information.

R: We have corrected this problems and deleted the words “Only for reviewers”. All data will be obvious and find on line. 

We wish the present version would satisfy the requirements for publication. Thank you greatly for your time spent in reviewing this work again.

Sincerely yours,

Yu Chen

School of physics science and electronics Central South University,

Changsha 410083, P.R. China

Reviewer 2 Report

Chen et al., demonstrate an interesting method to investigate the remote-excitation and remote-controlling of the localized surface plasmon resonance (LSPR) in a heterotype and hollow gold nanosheet (HGNS) using FDTD simulations. The heterotype HGNS contains an equilateral and hollow triangle in the center of a special hexagon, which forms a so-called hexagon-triangle (H-T) heterotype HGNS. By focussing the exciting incident laser on one of the vertexes of the center triangle, the LSPR could be achieved among other remote vertexes of the outer hexagon which is intriguing. It is observed that the peak wavelength and intensity of the LSPR depends sensitively on the factors such as the polarization of the incident light, the size and symmetry of the H-T 20 structure, to name a few. Based on the polar plots, the on-off switching of the LSPR coupled among four HGNS hotspots could be remote-controlled simply via only one polarized light, which exhibits great potential application in remote-excitation surface-enhanced Raman scattering (SERS) and other waveguide system. This research work is well-planned and executed. However, the discussion section is not comprehensive and need substantial revision. This interesting work may be considered for publication, provided the authors address the below mentioned comments.

1. The discussion section needs to be significantly improved.

2. The motivation behind this study is not explicit. Please provide adequate discussion on the same.

3. Authors should provide insights from fundamental aspects of LSPR, void and cavity and propagating plasmons. Additionally discussion on delocalized and localized plasmons observed in recent reports on soret colloids, Nd2O3-Ag, AgAu hybrids can be used to enhance the discussion.

4. Figure 1 a-h are in the 2-10 um resolution. A few high resolution images as well as images at lower magnification showing several such structures should be presented to provide discussion on the homogeneous characteristics.

5. The description/discussion on Figure 2-10 are minimal. Several insights from the photo-plasmonics point of view is missing although the experimental results are very promising for experimentalists in this domain. Please incorporate.

6. All the parameters utilized for the simulations should be explained in detail. It is instructive to present a schematic of the frameworks considered for the simulations.

7. The importance and relevance of the current research for future research in this direction, with perspectives needs to be detailed using relevant references: Micromachines 2023, 14(3), 668 ; Nanomaterials 2020, 10(9), 1667.

Needs some improvement.

Author Response

Response to Reviewer’s comments (ID: nanomaterials-2422324):

Thank you greatly for your time and effort spent in judging this work. We have carefully revised the manuscript according to the Reviewers’ comments. The revised parts have been marked in red in the new version.

Response to the reviewer 2:

  1. The discussion section needs to be significantly improved.

R: We have revised the discussion section carefully. The corrections in the revised version were marked using red words.

  1. The motivation behind this study is not explicit. Please provide adequate discussion on the same.

R: The main point in this article is focused on how dose LSPR and LSPR-coupling could be controlled remotely using one polarized light. This article on the whole has displayed the potential that how to use one polarized light to control LSPR and LSPR-coupling in a remote technique and found the optimized parameters that cover all coupling modes to light on/off of LSPR. In the revised version, we provided more descriptions on this main points. 

  1. Authors should provide insights from fundamental aspects of LSPR, void and cavity and propagating plasmons. Additionally discussion on delocalized and localized plasmons observed in recent reports on soret colloids, Nd2O3-Ag, AgAu hybrids can be used to enhance the discussion.

R: We have revised accordingly in the discussion section and add some references.

  1. Figure 1 a-h are in the 2-10 um resolution. A few high resolution images as well as images at lower magnification showing several such structures should be presented to provide discussion on the homogeneous characteristics.The description/discussion on Figure 2-10 are minimal. Several insights from the photo-plasmonics point of view is missing although the experimental results are very promising for experimentalists in this domain. Please incorporate.

R: Additional SEM images of solid and hollow Au nanosheet could be found in Figure 1. In fact, the thickness of solid Au nanosheet could be controlled smaller than 50 nm. However, the thickness of the as-synthesized hollow Au nanosheet is increased up to ~ 150 nm, which greatly influences its performance in remote-excitation of SERS. Thus, present FDTD theoretical studies is useful for the following researches if the fabrication technique could be improved that meets the basic requirement of the sheet thickness.

Figure 1. Additional SEM images of solid and hollow Au nanosheet. (a)-(c) Solid Au nanosheet at different magnification. (d) Hollow Au nanosheet. (e) EDX mapping images of (d). (f) EDX spectrum.

  1. All the parameters utilized for the simulations should be explained in detail. It is instructive to present a schematic of the frameworks considered for the simulations.

R: We have revised thoroughly the section of “Computational FDTD modelling details”, which describes the details of the parameters and their settings in the FDTD simulation. The revised parts have been marked in red in the new version. 

  1. The importance and relevance of the current research for future research in this direction, with perspectives needs to be detailed using relevant references: Micromachines 2023, 14(3), 668 ; Nanomaterials 2020, 10(9), 1667.

R: We have revised accordingly. The references have been added in the new version.

We wish the present version would satisfy the requirements for publication. Thank you greatly for your time spent in reviewing this work again.

Sincerely yours,

Yu Chen

School of physics science and electronics Central South University,

Changsha 410083,

P.R. China

Round 2

Reviewer 1 Report

I had got response on all related points. Although I had got a discrepancy in reply on the first point (waist radius are 2 μm, 5 μm and 8 μm, respectively, in the reply and 10 times smaller in the related Figure), but now this paper can be publishes as it is. My recommendation to prepare additional paper that uses the wide optical beam that corresponds the experimental situation. The current paper will be very helpful to explain the future modeling results.

Author Response

Response to Reviewer’s comments (ID: nanomaterials-2422324):

Dear Reviewers:

Thank you greatly for your time and effort spent in judging this work. We have carefully revised the manuscript according to the Reviewers’ comments. The revised parts have been marked in red in the new version.

Response to the reviewer #1:

  1. I had got response on all related points. Although I had got a discrepancy in reply on the first point (waist radius are 2 μm, 5 μm and 8 μm, respectively, in the reply and 10 times smaller in the related Figure), but now this paper can be publishes as it is. My recommendation to prepare additional paper that uses the wide optical beam that corresponds the experimental situation. The current paper will be very helpful to explain the future modeling results.

R: Thanks for the reviewer’s suggestions. We have carefully checked and improved again the English writing in the revised manuscript.

We wish the present version would satisfy the requirements for publication. Thank you greatly for your time spent in reviewing this work again.

Sincerely yours,

Yu Chen

School of physics science and electronics Central South University,

Changsha 410083,

P.R. China

Reviewer 2 Report

The authors claimed to have addressed the reviewer comments! However, appropriate revisions are not made. Especially, the references section is not comprehensive and authors should incorporate the comments suggested in the first round. This article may be re-considered after the authors have incorporated the reviewer's comments given in the first round.

Needs some more improvement.

Author Response

Response to Reviewer’s comments (ID: nanomaterials-2422324):

Dear Reviewers:

Thank you greatly for your time and effort spent in judging this work. We have carefully revised the manuscript according to the Reviewers’ comments. The revised parts have been marked in red in the new version.

Response to the reviewer #2:

  1. The authors claimed to have addressed the reviewer comments! However, appropriate revisions are not made. Especially, the references section is not comprehensive and authors should incorporate the comments suggested in the first round. This article may be re-considered after the authors have incorporated the reviewer's comments given in the first round.

R: The corrections include as follows:

  1. We revised Figure S1 and added some SEM images of the solid gold nanosheet in different magnifications.
  2. In the last version, an error was occurred on Endnote in the link of the references. Thus, we revised the references section again this time.
  3. According to the reviewer’s suggestions. We have carefully checked and improved again the English writing in the revised manuscript.

We wish the present version would satisfy the requirements for publication. Thank you greatly for your time spent in reviewing this work again.

Sincerely yours,

Yu Chen

School of physics science and electronics Central South University,

Changsha 410083,

P.R. China

Round 3

Reviewer 2 Report

authors addressed the reviewer comments.

needs some improvement